# ImagineNav: Prompting Vision-Language Models as Embodied Navigator through Scene Imagination

**Xinxin Zhao**\*, **Wenzhe Cai**\*, **Likun Tang**, **Teng Wang**†
School of Automation, Southeast University
{xinxin␣zhao, wz␣cai, lktang, wangteng}@seu.edu.cn

## Abstract

Visual navigation is an essential skill for home-assistance robots, providing the object-searching ability to accomplish long-horizon daily tasks. Many recent approaches use Large Language Models (LLMs) for commonsense inference to improve exploration efficiency. However, the planning process of LLMs is limited within texts and it is difficult to represent the spatial occupancy and geometry layout only by texts. Both are important for making rational navigation decisions. In this work, we seek to unleash the spatial perception and planning ability of Vision-Language Models (VLMs), and explore whether the VLM, with only on-board camera captured RGB/RGB-D stream inputs, can efficiently finish the visual navigation tasks in a mapless manner. We achieve this by developing the imagination-powered navigation framework **ImagineNav**, which imagines the future observation images at valuable robot views and translates the complex navigation planning process into a rather simple best-view image selection problem for VLM. To generate appropriate candidate robot views for imagination, we introduce the **Where2Imagine** module, which is distilled to align with human navigation habits. Finally, to reach the VLM preferred views, an off-the-shelf point-goal navigation policy is utilized. Empirical experiments on the challenging open-vocabulary object navigation benchmarks demonstrates the superiority of our proposed system.

## 1 Introduction

A useful home-assistant robot should be able to search for different kinds of objects without telling it the exact 3D object coordinates for completing our human instructions. As our human always buying and bringing new goods back home, the robot's object searching capability should not be limited in a closed-set of categories. Researchers often refer this problem as open-vocabulary object navigation task. Recently, as the emergence of the foundation models, including capable vision models (Radford et al., 2021; He et al., 2022; Zhou et al., 2022; Cheng et al., 2024; Kirillov et al., 2023; Wu et al., 2024a; Liu et al., 2023), large language models (LLMs) and vision-language models (VLMs) (Brown et al., 2020; Chowdhery et al., 2023; Zhang et al., 2022; Touvron et al., 2023; Achiam et al., 2023; Team et al., 2023; Liu et al., 2024; Chen et al., 2024; Dai et al., 2023; Gao et al., 2023), building an agent that can accomplish the open-vocabulary object navigation becomes possible. A popular framework, as shown in Figure 1, uses modular approach to deal with this problem, which often composes of four components: A real-time mapping and segmentation module to perceive robot surrounding environments. A template-based translation module to compress the semantic map information into texts. A LLM-based module to understand the textual information from the previous step and make a plan in texts. Finally, a path-planning module which project the reasoning result from LLM back to the map and plan a collision-free path navigating towards it.

Although such a pipeline achieves great success in recent years (Zhou et al., 2023; Kuang et al., 2024; Wu et al., 2024b; Zhang et al., 2024a; Shah et al., 2023; Yu et al., 2023a; Loo et al., 2024), these complex cascaded systems have several limitations. Firstly, both the depth camera and the robot localization module can suffer from perception error, especially for long-range depth estimation, and this can make the mapping process inaccurate. Secondly, online object detection and segmentation are required to augment spatial maps with semantic labels and prepare for LLM's

---

\*Equal Contribution, †Corresponding Author: Teng Wang.

reasoning input. This increases the computational burden for the robots. Thirdly, although the semantic information stored on the map can be easily expressed by text (e.g., list the categories of the observed objects), such pure text prompts have difficultly in explicitly describing the geometry information and object details in the map, making it difficult and ambiguous for LLMs to infer the best navigation plan.

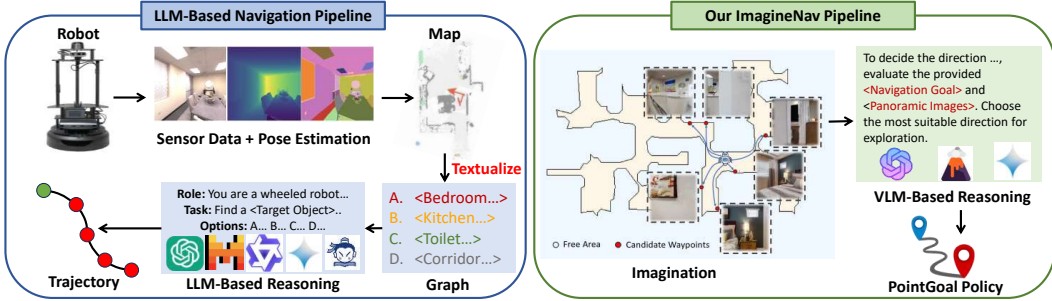

Figure 1: The comparison between the conventional LLM-based navigation pipeline and our ImagineNav pipeline. The traditional LLM-based navigation framework, illustrated on the left, relies on intricate sensor data processing and pose estimation for map creation, followed by LLM-driven reasoning to decide the exploration direction. Instead, our ImagineNav directly translates the long-horizon object goal navigation task into a sequence of best-view image selection tasks for VLM, which avoids the latency and compounding error in the traditional cascaded methods.

In this work, we try to explore whether it is possible to circumvent the complicated and fragile *mapping→ translation→ planning* framework, but develop a visual navigation approach with only raw RGB/RGB-D observations and pre-trained VLMs. Our proposed method - **ImagineNav** seeks to maximize the capabilities of VLMs in multimodal understanding and reasoning, and make the VLMs become an efficient embodied navigation agent. As most VLMs cannot understand the continuous physical world, it is infeasible to directly ask VLMs to generate navigable 3D waypoints. Instead, we translate the visual navigation problem into an imagination-powered best-view image selection task and let the VLMs select. To generate appropriate options for VLMs to choose from, we propose the **Where2Imagine** model which distills the human indoor navigation habits and generates future 3D navigation waypoints where a human might navigate based on the current visual observation. Such 3D navigation waypoints indicate relative poses with respect to the current frame and can be easily translated into new observation images using novel view synthesis (NVS) models. Afterwards, the VLMs only need to select the best imagined observation that is mostly related to the target object and drive the robot to follow the corresponding point-goal navigation trajectory. The above pose-aware imagination-and-selection capability allows the Object Goal Navigation (ObjectNav) task to be decomposed into a sequence of point-goal sub-tasks, facilitating the creation of collision-free navigation trajectories. Experimental results on standard benchmarks demonstrate the superiority of our ImagineNav over previous methods in open-vocabulary object navigation. In summary, our contributions are:

- We propose a mapless navigation approach ImagineNav. It leverages the imagination to generate image observations at potential future 3D waypoints as the VLMs' visual prompts, grounding the VLMs to become efficient navigation agents without any fine-tuning.

- We design a task-oriented model Where2Imagine to understand human navigation habits. This model is crucial to bridge the task-agnostic high-level VLM planners and the low-level navigation policies.

- Our ImagineNav increases success rate by a large margin of 15.1% and 10% respectively on HM3D (Ramakrishnan et al., 2021) and HSSD (Khanna et al., 2023). We also provide a detailed ablation analysis to help understand the important conclusions in our framework.

## 2 RELATED WORKS

### 2.1 LARGE MODELS FOR ROBOTIC PLANNING

Large-scale models pre-trained on extensive internet data have demonstrated formidable zero-shot reasoning capabilities in tasks such as planning (Huang et al., 2022), code generation (Liang et al.,

2023; Huang et al., 2023b), and solving science questions (Lewkowycz et al., 2022). The in-context learning capability of LLMs allows them to be applied to robotic task planning. Some methods (Liang et al., 2023; Huang et al., 2023b; Ahn et al., 2022) leverage LLMs to decompose tasks into subtasks, enhancing execution efficiency. Cap (Liang et al., 2023) generates robotic policy code directly from example language commands, enabling autonomous control and task execution based on natural language instructions. Instruct2Act (Huang et al., 2023b) combines LLM with foundational models (e.g., SAM and CLIP), reducing error rates in complex task execution, while SayCan (Ahn et al., 2022) combines LLM task planning with the feasibility of physical skills using pre-trained value functions, generating actionable plans for robots. However, one limitation of LLMs is their difficulty in embedding the robot's state directly into the planning process. To address this, many studies have turned to VLMs as alternatives. For instance, ViLA (Lin et al., 2024) significantly improves performance on multimodal tasks without compromising text capabilities by systematically exploring VLM pretraining design choices. CoPa (Huang et al., 2024) incorporates commonsense knowledge from VLMs, proposing a coarse-to-fine task-oriented grasping and task-aware motion planning approach. PIVOT (Nasiriany et al., 2024) transforms tasks into iteratively optimized visual question-answering problems via a refinement process. Socratic model (Zeng et al., 2022) integrates multiple pretrained large models (e.g., VLMs, LLMs, and audio models) in a modular fashion to enable reasoning and task execution through language-based interaction. These methods employ a set-of-examples (SOE) approach to guide VLM selection. We propose a new decision-making paradigm based on imagined imagery, wherein decisions are made on imaginations, enabling more nuanced, context-aware interactions that better harness VLMs' spatial perception capabilities.

## 2.2 OPEN-VOCABULARY OBJECT NAVIGATION

ObjectNav requires the robot to navigate toward a specific target category in unseen environments. Although previous works (Yadav et al., 2022; Ramrakhya et al., 2023; Chaplot et al., 2020; Ramakrishnan et al., 2022) can achieve high success rate in widely accepted benchmarks (e.g., habitat-challenge (Yadav et al., 2023)), most approaches are limited within a pre-defined object list, which is contradictory to the open-vocabulary real world. Therefore, many researchers start to discuss the open-vocabulary object navigation problem. End-to-end methods try to make use of compact multimodal features space (e.g., CLIP (Radford et al., 2021)) for grounding text knowledge into visual navigation problem but achieved limited performance (Khandelwal et al., 2022; Gadre et al., 2023; Majumdar et al., 2022). Instead, modular-based approaches (Huang et al., 2023a; Zhou et al., 2023; Achiam et al., 2023) typically necessitate the use of sensors for localization and mapping, high-level planning, and low-level control. These methods rely heavily on high-precision sensors for accurate self-localization and real-time map construction. Our approach introduces an imagination-based, mapless navigation framework. This framework circumvents the need for extensive training by transforming the complex process of navigation planning into a selection problem based solely on RGB inputs.

## 2.3 IMAGINATION-BASED NAVIGATION

Recent methods (Zhai & Wang, 2022; Ramakrishnan et al., 2022; Zhu et al., 2022) have employed supervised learning to learn target-related functions in order to address the subtask of 'Where to look?' in navigation, specifically focusing on predicting the localization of target. These methods predict the absolute coordinates (Zhai & Wang, 2022) of the target, the shortest distance to target (Ramakrishnan et al., 2022), or the nearest boundary (Zhu et al., 2022) based on local maps. Instead, several studies (Ramakrishnan et al., 2020; Georgakis et al., 2022; Liang et al., 2021; Zhang et al., 2024b) have proposed various approaches to enhance the prediction of unobserved regions. For instance, (Ramakrishnan et al., 2020) introduced occupancy anticipation, where the agent infers an occupancy map based on RGB-D inputs. The L2M framework was introduced (Georgakis et al., 2022), consisting of a two-stage segmentation model that generates a semantic map beyond the agent's field of view and selects long-term goals based on the uncertainty of predictions. SS-CNav algorithm (Liang et al., 2021) leverages semantic scene completion and confidence maps to infer the environment and guide navigation decisions. A self-supervised generative map (SGM) is proposed (Zhang et al., 2024b), which employs self-supervised approach to continually generate unobserved regions in the local map and predict the target's location. These methods primarily predict unobserved regions in top-down maps derived from egocentric RGB-D projections. In contrast,

Our method operates in the RGB space to perform guided-imagination. We utilize a compact model aligned with human navigation habits to generate new viewpoints, followed by the imagination process to produce corresponding visual representations.

# 3 METHODOLOGY

The open-vocabulary object navigation task requires the agent to locate an untrained target object instance in an unknown environment. At the start of each episode, the agent is spawned at a random position in an unfamiliar environment without any prior knowledge of the layout. The task is to find a target object $g_i$, which can belong to any category in an open-vocabulary setting. At each time step $t$, the agent receives an egocentric panorama view $I_t$, divided into 6 separate views, each represented by an RGB image $I_{t,i}$ accompanied by its depth map $D_{t,i}$. The discrete action space consists of the following commands: {Stop, MoveAhead, TurnLeft, TurnRight, LookUp, LookDown}. The 'MoveAhead' action moves the agent forward by 0.25m, while the rotational actions 'TurnLeft' and 'TurnRight' rotate the agent by 30 degrees. The task is considered successful if the agent reaches the target object with a geodesic distance smaller than a defined threshold (e.g., 1m) and executes the 'Stop' command within a fixed number of steps. Each episode has a maximum limit of 500 steps.

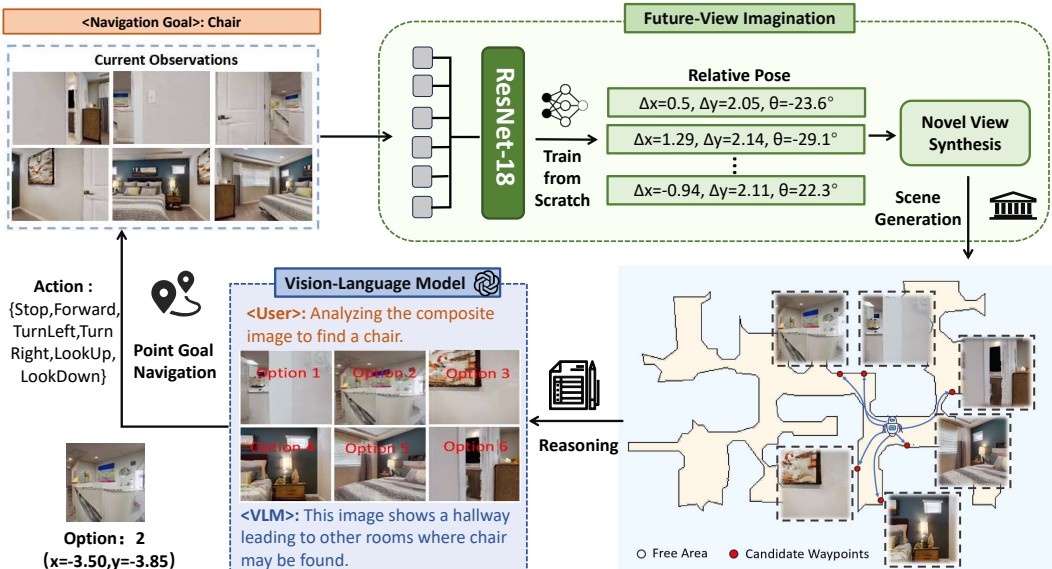

Figure 2: The overall pipeline of our mapless, open-vocabulary navigation framework. At each iteration, the agent captures a panoramic view of its surroundings. In the Imagination Module, the trained Where2Imagine module couples with novel view synthesis model to generate novel scene views. Guided by prompt templates, the VLM engages in target-oriented inference. Subsequently, the system executes the PointNav policy to determine the next navigational action. The above imagination, reasoning and planning procedure iterates until the target is reached.

## 3.1 IMAGINENAV FRAMEWORK

This section provides an overview of the imagination-based open-vocabulary object navigation framework (ImagineNav). As depicted in Figure 2, the agent initially employs the Where2Imagine module to generate candidate locations for imagination based on the current observations. Subsequently, the visual observations at these locations are imagined by a NVS model. Utilizing the generated images, which are annotated with option labels, the agent leverages a multimodal large-scale model to assess both the spatial and semantic information of each scene, enabling the selection of a more efficient exploration direction. Specifically, the VLM is employed to reason over the imagined images from six different views using prompts, selecting the optimal waypoint. Finally, the agent executes low-level point navigation strategy to reach the corresponding sub-goal. This process iterates, where each new observation serves as input for further imagination, reasoning, and

navigation, until the agent successfully identifies an instance of the target object. Consequently, the ObjectNav task is reduced to a sequence of simpler point-to-point navigation subtasks. Since our ImagineNav does not require any training on object-oriented data for reasoning and planning, it is open-vocabulary and can zero-shot generalize to novel semantic targets. Sec 3.2 introduces the imagination module, Sec 3.3 explains the use of the advanced planner VLM, and Sec 3.4 describes our underlying point navigation strategy.

## 3.2 FUTURE-VIEW IMAGINATION

To better leverage the spatial perception and reasoning capabilities of VLMs for open-vocabulary object navigation in unknown environments, we propose an future-view imagination model, which is composed by Where2Imagine module and a NVS model. As shown in Figure 2, Where2Imagine predicts the relative pose $(\Delta x, \Delta y, \theta)$ of the potential next navigation waypoint based on the current RGB observation, where $\Delta x$ denotes lateral displacement, $\Delta y$ represents longitudinal displacement, and $\theta$ refers to changes in the camera's viewing angle. Subsequently, a novel view image is generated based on the predicted relative pose. There are several advanced methods to achieve this, such as framing the task as a few-shot 3D rendering problem (Sargent et al., 2024; Wimbauer et al., 2023; Cao & de Charette, 2023) or utilizing generative models (e.g., diffusion models) for image synthesis (Yu et al., 2024; Tseng et al., 2023; Yu et al., 2023b). In this work, we employ a pretrained diffusion model (Yu et al., 2024), which generates new view images by taking the current image and the predicted relative pose as inputs, enabling high-quality viewpoint transformation.

Specifically, we trained a ResNet-18 from scratch for the relative waypoint prediction task, with training data sourced from the Habitat-Web project (Ramrakhya et al., 2022). Habitat-Web collects remote user demonstrations of virtual robot operations through a virtual teleoperation system on the Amazon Mechanical Turk platform, including 80k ObjectNav and 12k Pick&Place demonstrations. Humans in these demonstrations typically opt for directions towards important semantic cues (e.g., doors) that facilitate exploration. These navigation preferences serve as a valuable basis for determining potential waypoints, thereby enhancing the rationality and safety of the navigation strategy. We transformed the human demonstration trajectories into a paired dataset $(I_t, P_{t+T})$, where $I_t$ represents the RGB image at frame $t$, and $P_{t+T}$ denotes the relative pose $(\Delta x, \Delta y, \theta)$ at frame $t + T$ with respect to frame $t$. To facilitate the learning of the network, we filtered out images with a depth threshold of less than 0.3, as these images generally lack rich semantic information (e.g., a plain wall). Due to the limitations of the NVS model in generating high-quality images for large perspective shifts (e.g., 120°,180°,240°) based solely on a single input image, we restricted the training data to instances where the angular deviation falls within ±30°. Through the Where2Imagine module, our imagination model aligns with human navigation habits.

## 3.3 HIGH-LEVEL PLANNING

High-level planning module leverages the spatial awareness and common-sense reasoning capabilities of the VLM to select the direction most likely to locate the navigation target. We prefer GPT-4o-mini as the high-level planner because it offers a balance between the reasoning capabilities and practical efficiency. Compared to larger models (e.g., GPT-4o), GPT-4o-mini is lightweight and cost-effective. Its smaller size ensures faster inference, allowing the system to make timely decisions in dynamic environments. To assist GPT-4o-mini in decision making, we designed a simple prompt template, requiring the VLM to summarize its choice in a JSON format containing { 'Reason', 'Choice' }. This format allows for a clear understanding of the VLM's reasoning process. As illustrated in Figure 3, the VLM receives the synthesized observations at future navigation waypoints and the navigation goal as inputs. Based on the prompt, 'Your choice should first be based on discovering navigation targets, followed by the potential of unexplored areas...' , the VLM analyzes each image's semantic information, selects the optimal exploration direction, and returns the answer in the specified format. By providing the imagined observations as visual prompts to VLM, our ImagineNav offers significant advantages in spatial reasoning and decision-making processes. Firstly, the VLMs are more skilled at handling multiple-choice decision tasks compared to 3D geometry question answering (i.e., directly inferring the 3D coordinates of next waypoints). Moreover, the introduction of imagination enhances the decision-making capability of the VLM by supplementing detailed information about distant or visually unclear objects. The advanced planning module proposes new sub-goals after the low-level controller completes navigation to the designated target.

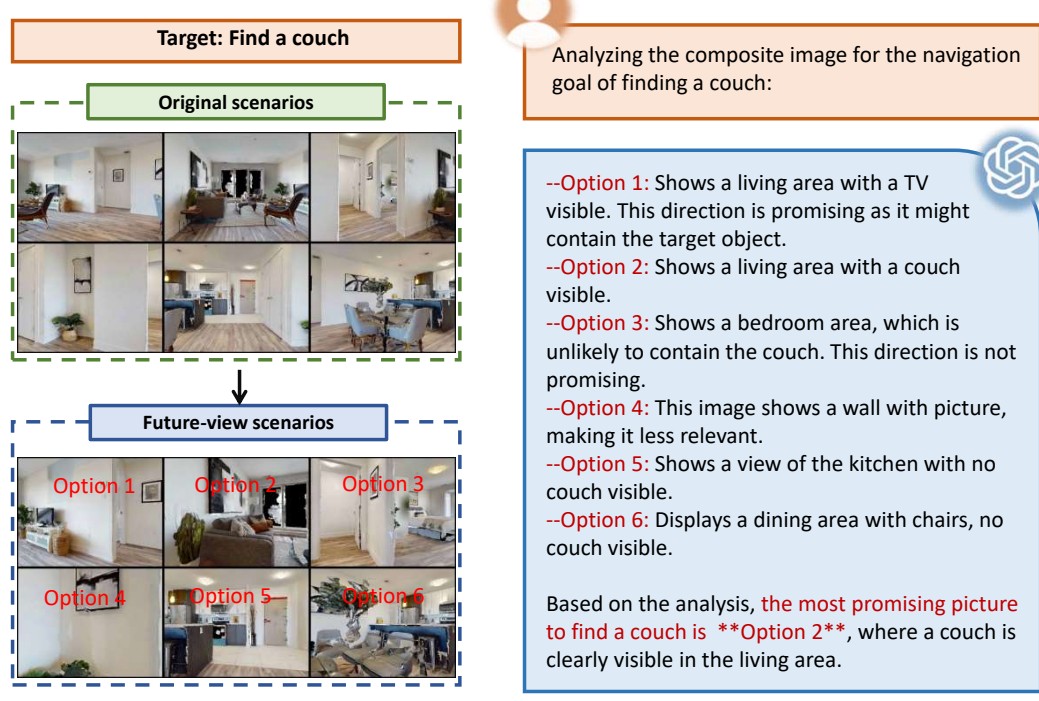

Figure 3: An example of the VLM analysis. By examining different future-view scenarios, the VLM pinpoints the direction most likely to incorporate the target object couch.

### 3.4 CONTROLLER

After the high-level planner provides the navigation points, the low-level controller executes Point Goal Navigation (PointNav) strategy to achieve these targets. Unlike ObjectNav, PointNav (turn to $\Delta x$, $\Delta y$) does not rely on semantic information from the environment but is instead driven solely by spatial perception. Currently, there are many methods available for achieving PointNav (Yang et al., 2023; Roth et al., 2024; Wijmans et al., 2022; Liang et al., 2024). To determine the execution actions at each step of the PointNav process, we use Variable Experience Rollout (VER) (Wijmans et al., 2022) as our underlying goal navigation strategy. VER combines the advantages of synchronous and asynchronous reinforcement learning, improving training efficiency and sample utilization in PointNav tasks, thereby enabling the agent to demonstrate stronger adaptability and generalization capabilities in complex environments.

## 4 EXPERIMENTS

### 4.1 EXPERIMENTAL SETUP AND METRICS

We evaluate the effectiveness and navigation efficiency of our proposed method using the Habitat v3.0 simulator (Puig et al., 2023) on two standard ObjectNav datasets: HM3D (Ramakrishnan et al., 2021) and HSSD (Khanna et al., 2023). The HM3D dataset offers high-fidelity reconstructions of 20 entire buildings, including 80 training scenes and 20 validation scenes. The HSSD dataset provides 40 high-quality synthetic scenes, comprising 110 training scenes and 40 validation scenes. The experimental setup follows the ObjectNav-challenge-2023 (Yadav et al., 2023). For the data collection of the Where2Imagine module, we leveraged human demonstration trajectories from the MP3D (Chang et al., 2017) dataset within the habitat-web project with the camera height 0.88m and horizontal field of view (HFOV) of 79°. We report the performance in terms of Success Rate (SR), defined as the proportion of episodes where the agent's distance to the target object is less than 1m after executing the STOP action, and *SPL* (Anderson et al., 2018), Success weighted by path length,

$SPL = \frac{1}{N} \sum_{i=1}^{N} S_i \left( \frac{\ell_i}{\max(p_i, \ell_i)} \right)$, where $S_i$ be a binary success indicator in episode $i$, $p_i$ is the agent path length and $\ell_i$ is the GT path length.

## 4.2 BASELINES

We conducted a comparative analysis of non-zero-shot and zero-shot ObjectNav methods to substantiate our proposed approach. Yamauchi (Topiwala et al., 2018) pioneered a frontier-based exploration strategy, emphasizing the boundaries between explored and unexplored regions. SemExp (Chaplot et al., 2020) advanced this concept by implementing goal-directed semantic exploration through the construction of semantic maps. In addition, we examined non-mapping closed-set object navigation baselines, including those based on imitation learning (Ramrakhya et al., 2022) and visual representation learning (Yadav et al., 2022).

For zero-shot object navigation, we consider several mapping-based baselines (Zhou et al., 2023; Wu et al., 2024b; Yokoyama et al., 2023) that integrate commonsense knowledge and semantic information to facilitate direct navigation toward target objects, leveraging semantic comprehension from pre-trained large models to aid in navigation. Furthermore, we explored RGB-based non-mapping navigation baselines: ZSON (Majumdar et al., 2022), which employs the CLIP (Radford et al., 2021) model to embed target images and object goals within a unified semantic space, thereby training a semantic goal-navigation agent, and PixNav (Cai et al., 2023), which utilizes pixel-level goal guidance, enabling pixel navigation through the use of VLMs and LLMs.

## 4.3 COMPARISON WITH PRIOR WORK

Table 1 presents a comparative analysis of the proposed ImagineNav against prior research efforts. On the HM3D dataset, ImagineNav achieves a success rate of 53.0% and a SPL of 23.8%, significantly outperforming most of the methods especially at success rate. Moreover, ImagineNav achieves the highest success rate and SPL on the HSSD dataset. Particularly, in open-vocabulary navigation tasks, our mapless ImagineNav even outperforms the best-performing map-based method VLFM (Yokoyama et al., 2023) by 0.5% at success rate. The above observations indicate that our ImagineNav demonstrates outstanding navigation performance across various settings, while maintaining low storage and computational complexities. Furthermore, since the pretrained NVS is directly employed without finetunned on the HM3D and HSSD datasets, we see a disparity between the quality of images generated by the NVS model and real images, limiting the capability of our model to some extent. To explore the upper limits of our framework, we instead use real panoramic images—specifically, the observation at the pose predicted by the Where2Imagine module—as visual prompts for the VLM model. Notably, both the success rate and SPL exhibit obvious improvements, obtaining 62.0% and 59.0% at success rate respectively on H3MD and HSSD benchmarks, which further demonstrates the superiority of our imagination-based navigation framework.

Table 1: **ImagineNav: Comparison with previous work.** The Where2Imagine model with T=11, utilizing ResNet-18 trained from scratch and GPT-4o-mini as the VLM, was evaluated over 200 epochs on the HM3D and HSSD datasets. ImagineNav uses NVS model to generate novel view images, while ImagineNav-Oracle uses real images of the candidate points.

| Method | Open-Vocabulary | Mapless | HM3D | | HSSD | |
|---|---|---|---|---|---|---|
| | | | Success Rate | SPL | Success Rate | SPL |
| FBE (Topiwala et al., 2018) | ✗ | ✗ | 33.7 | 15.3 | 36.0 | 17.7 |
| SemExp (Chaplot et al., 2020) | ✗ | ✗ | 37.9 | 18.8 | - | - |
| Habitat-Web (Ramrakhya et al., 2022) | ✗ | ✓ | 41.5 | 16.0 | - | - |
| OVRL (Yadav et al., 2022) | ✗ | ✓ | 62.0 | 26.8 | - | - |
| ESC (Zhou et al., 2023) | ✓ | ✗ | 39.2 | 22.3 | - | - |
| VoroNav (Wu et al., 2024b) | ✓ | ✗ | 42.0 | 26.0 | 41.0 | 23.2 |
| VLFM (Yokoyama et al., 2023) | ✓ | ✗ | 52.5 | 30.4 | - | - |
| ZSON (Majumdar et al., 2022) | ✓ | ✓ | 25.5 | 12.6 | - | - |
| PixNav (Cai et al., 2023) | ✓ | ✓ | 37.9 | 20.5 | - | - |
| **ImagineNav** | ✓ | ✓ | **53.0** | **23.8** | **51.0** | **24.9** |
| **ImagineNav-Oracle** | ✓ | ✓ | **62.0** | **31.1** | **59.0** | **27.0** |

## 4.4 ABLATION STUDY ON MAIN COMPONENTS

We conducted an ablation study on main components of the imagination module to demonstrate their effectiveness, with each variant evaluated for 100 epochs. As shown in Table 2, we increase success rate from 43.0 to 55.0 by utilizing future imaginations as visual prompt of the VLM for deciding exploration direction. Please note that at row #2 without Where2Imagine, we simply generate future views at the six locations, which are two meters away from current observations along their respective observation orientations. This improvement suggests the superiority of future imagination in facilitating VLM's reasoning. Such improvements can be attributed to the greater semantic disparity between different imaginations, as illustrated in Figure

Table 2: **ImagineNav: ablation study on the imagination module**. 'Imagination' refers to whether the future imaginations are used as visual prompts of the VLM. When it is removed, we feed current observations into VLM for deciding the best exploration direction, and set the next waypoint 2 meters away from the current location along the direction. Here, the distance of 2 meters is considered as it is comparable to that generated by T=11. 'NVS' indicates whether the image is captured from a real environment or synthesized via the NVS model.

| Imagination | Where2Imagine | NVS | HM3D | |
| --- | --- | --- | --- | --- |
| | | | Success Rate | SPL |
| ✗ | ✗ | Oracle | 43.0 | 24.7 |
| ✓ | ✗ | Oracle | 55.0 | 27.6 |
| ✓ | ✓ | Oracle | 64.0 | 28.3 |
| ✓ | ✗ | PolyOculus | 49.0 | 23.3 |
| ✓ | ✓ | PolyOculus | 56.0 | 24.3 |

4. Further incorporating Where2Image improve success rate from 55.0 to 64.0, and from 49.0 to 56.0 under settings of 'NVS' and 'w/o NVS', respectively. As mentioned above, the capability of our ImagineNav is limited by the performance of the off-the-shelf NVS model (Yu et al., 2024) to some extent as evidenced by comparing rows #2 with #4 and rows #3 with #5. Nevertheless, the incorporation of Where2Imagine partially mitigates the adverse effects of the NVS model.

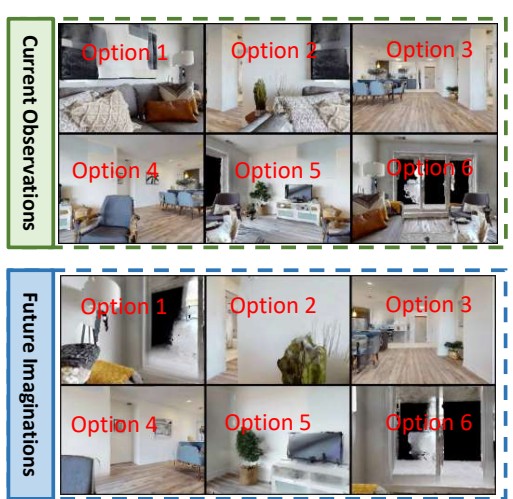

Figure 4: Visualization of the synthesized image observations at future navigation waypoints predicted by the imagination module. It can be seen that there exists drastic semantic disparity between different imaginations. In contrast, the semantic information is relatively consistent across different current observations. The varying semantics across different future views highlight the advantages of the imagination module in enhancing the VLM's decision-making capabilities.

## 4.5 ANALYSIS OF SUCCESSFUL AND FAILED TRAJECTORIES

Figure 5 illustrates that our method achieves efficient path planning and navigation across different targets. Especially, as shown in the top middle of Figure 5, the agent needs to navigate through multiple rooms. The complex environment increases the risk of losing direction, but our ImagineNav successfully infers the optimal path and finds the target, demonstrating its ability to handle long-distance and multi-room scenarios. We also present some failure examples at the bottom of Figure

5. We identified three key factors contributing to these navigation failures. First, some object instances are neglected for marking by the simulator, and therefore a successfully trajectory is wrongly considered as a failure (a.k.a. false failure) as shown in the bottom left of Figure 5. Second, the synthesized image from the NVS does not align with the real observation, such as creating objects that are not present in the real scene as shown in the middle of Figure 5, which causes the VLM to make incorrect inference. Moreover, the lack of historical information make the agent easily trapped in local optima, thereby limiting its performance in long-term navigationas shown in the bottom right of Figure 5.

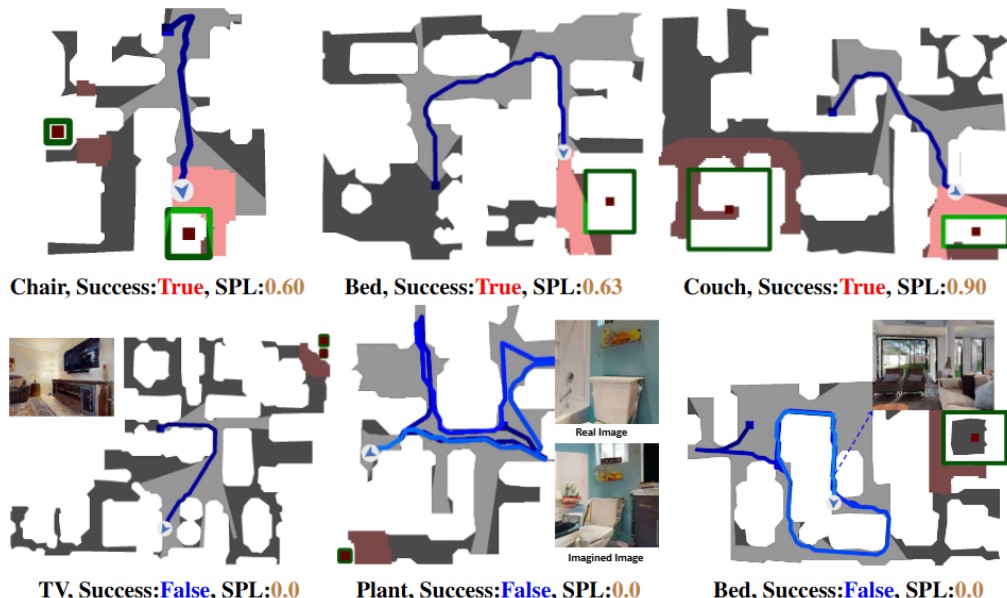

Figure 5: Visualization of the navigation trajectory. The top and bottom rows respectively show the complete top-down trajectories of successful and unsuccessful examples.

## 4.6    ANALYSIS OF WHERE2IMAGINE MODULE

We explore the impact of the sampling step T on the final navigation performance by varying T from 8 to 15. For each T, we re-generate the labeled image data and re-train the ResNet-18 for relative pose prediction. Each variant was tested for 100 epochs under conditions where the agent had access to real panoramic observations. As shown in Table 3, the best success rate and SPL are obtained when T is set to 11. Furthermore, we visualize several navigational trajectories under different values of T in Figure 6 to facilitate explanation. As can be seen, when T is relatively small (i,e., 8), the agent is easily trapped as marked by red square, since it mainly resorts to local semantic information for inferring its exploration direction, making it susceptible to converging on suboptimal local solutions.

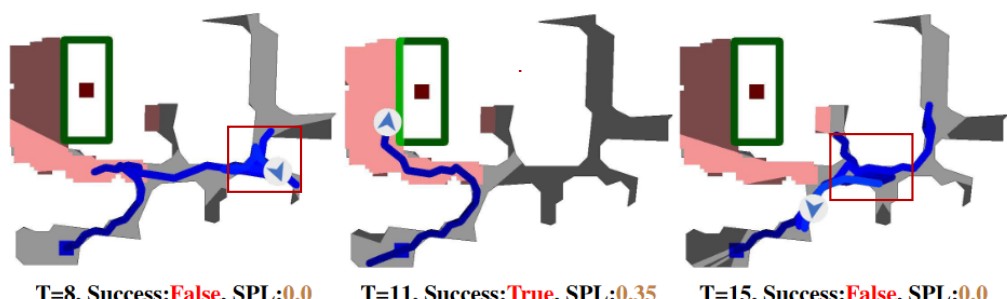

Figure 6: Comparison of trajectories at different sampling steps T. This image presents a top-down view of the entire trajectory as the agent searches for the target (a chair). The red box highlights the situation where the agent encounters a local trap during navigation.

Conversely, when T is excessively large, although the agent has access to more distant information, it is prone to miss some critical intermediate semantics which are closely related to target and are worth exploring, leading to erroneous long-range decisions, particularly in intricate environments. However, an optimally calibrated T can strike a delicate balance between exploration and perception, thereby facilitating to obtain impressive navigation performance.

We compared different backbones to evaluate their impacts on both the relative waypoint prediction and final navigation performances. Specifically, we modified the final output layers of the ResNet-18 and ViT to fit our dataset, allowing parameter updates during training. In contrast, DINOv2 and MAE models were connected to a five-layer MLP, with only the MLP parameters trained while freezing the backbone. The experiments were conducted on real RGB observations without resorting to the NVS model. Each variant was tested for 100 epochs. The results in Table 4 show that the ResNet-18, when trained from scratch, achieves the best performances in both relative wayoint prediction and ObjectNav while featuring a more lightweight architecture. Furthermore, the large performance gap between different backnones suggests the importance of the Where2Imagine module, indicating the usefulness of learning from human demonstrations. Please note that we observe a slight inconsistency between waypoint prediction loss and navigation success rate under backnones of DINOv2 and MAE. This might be because the waypoint prediction is evaluated on the testing data from MP3D dataset while the navigation is performed on HM3D. The differences between the datasets result in a certain degree of variability.

Table 3: **Where2Imagine:** the influence of sampling step **T** on navigation performance.

| T | HM3D | |
|---|---|---|
| | Success Rate | SPL |
| **8** | 51.0 | 25.1 |
| **10** | 64.0 | 29.4 |
| **11** | 64.0 | 28.3 |
| **12** | 59.0 | 30.0 |
| **15** | 59.0 | 26.8 |

Table 4: **Where2Imagine:** the impact of different **backbones**. Loss refers to the test loss of Where2Imagine. TFS: training from scratch, FT: fine-tuning.

| Backbone | Params | Flops | Loss | HM3D | |
|---|---|---|---|---|---|
| | | | | Success Rate | SPL |
| **ResNet-18 (TFS)** | 11.4M | 1.8G | 0.12 | 64.0 | 28.3 |
| **ResNet-18 (FT)** | 11.4M | 1.8G | 0.24 | 61.0 | 29.7 |
| **ViT (TFS)** | 86.0M | 16.9G | 0.22 | 61.0 | 29.7 |
| **ViT (FT)** | 86.0M | 16.9G | 0.23 | 58.0 | 31.0 |
| **DINOv2** | 22.6M | 5.5G | 0.22 | 58.0 | 27.9 |
| **MAE** | 87.1M | 4.4G | 0.20 | 57.0 | 26.5 |

## 4.7 ANALYSIS OF VLM PLANNER

We conducted a comparative evaluation of the effects of different VLMs on navigation performance, as detailed in Table 5. The experiments used real RGB without NVS model. The results demonstrate that GPT-4o-mini and GPT-4-Turbo exhibit negligible differences in success rate and SPL metrics, while showing a marked advantage over LLaVa, underscoring the significant role of advanced model reasoning capabilities in influencing experimental outcomes. Moreover, for models of the same architecture, it is possible to opt for more cost-effective variants without compromising navigation performance, thus enabling more resource-efficient and time-saving.

Table 5: Effect of different VLM.

| VLM | HM3D | |
|---|---|---|
| | Success Rate | SPL |
| **LLaVa** | 44.0 | 21.6 |
| **GPT-4-Turbo** | 63.0 | 29.4 |
| **GPT-4o-mini** | 64.0 | 28.3 |

## 5 CONCLUSION

We propose the ImagineNav framework, a mapless, open-vocabulary object navigation approach leveraging VLM. By incorporating imagination mechanism, the system effectively predicts future waypoints, transforming traditional navigation planning into a visual selection task for the VLM. Ablation studies highlight ImagineNav's strong potential for long-horizon navigation. Moving forward, we aim to enhance the quality of viewpoint generation and optimize the use of historical memory to further improve navigation performance and robustness.

## 6 ACKNOWLEDGEMENT

This work was supported by the National Natural Science Foundation of China (Grant No. 62273093).

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
