# A APPENDIX

## A.1 DECISION-MAKING DETAILS OF VISION-LANGUAGE MODELS

The process of inputting data into the VLM is as follows: First, the agent acquires an RGB observation based on its current pose and the Where2Imagine module predicts a relative pose based on the current observation. The RGB observation and the relative pose predicted by Where2Imagine are jointly input into the NVS to generate a new view image. This process is repeated at intervals of 60°, ultimately producing six novel view images, which are stitched together to serve as input to the VLM. In the ablation experiments, the VLM inputs for different variants differ in whether or not Where2Imagine is used for relative pose prediction and novel view generation, but at the start of each cycle, the initial input to the model is the RGB observation from the agent's current position at 60° intervals. Figure 7 shows the complete prompts and responses of the VLM.

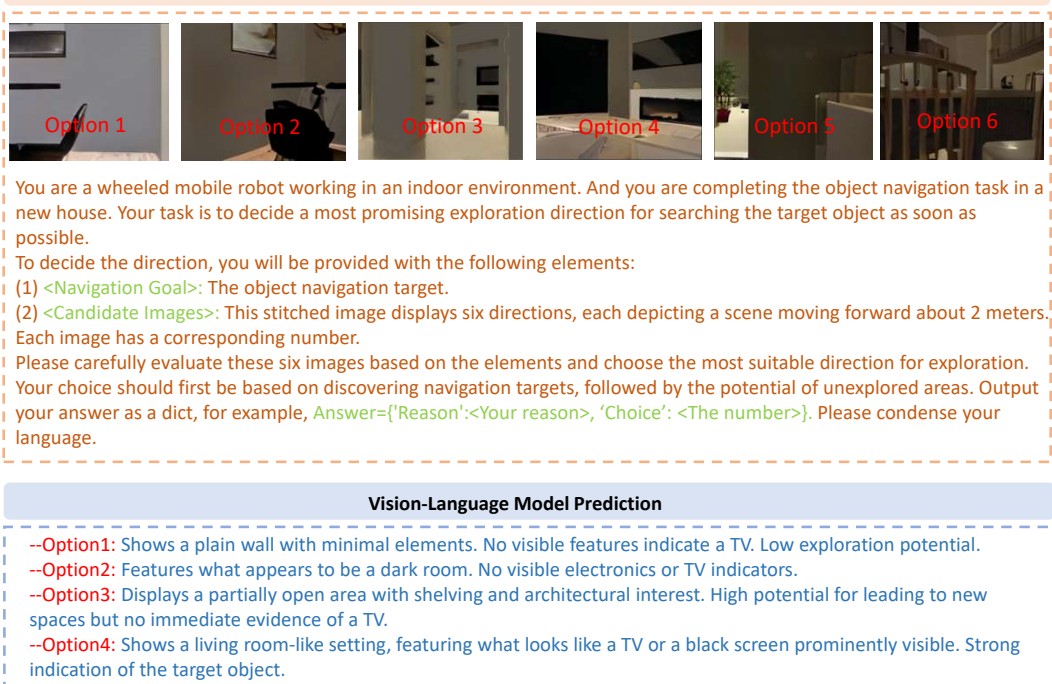

Figure 7: Complete prompt input and decision output of vision-language model.

## A.2 COMPARISON BETWEEN THE WHERE2IMAGINE MODULE AND UNIFORM SAMPLING

As shown in Figure 8, the prediction points of the Where2Imagine module are more concentrated in walkable areas, and the predicted views are richer in information, covering more structural and semantic details such as furniture, doors, and windows within the room. In contrast, the r=2.0m sampling points are uniformly distributed based on a fixed radius, without considering obstacles or scene information. Where2Imagine's prediction tendency not only aligns better with realistic navigable paths but also effectively guides the agent toward areas with higher information density, improving its environmental perception efficiency.

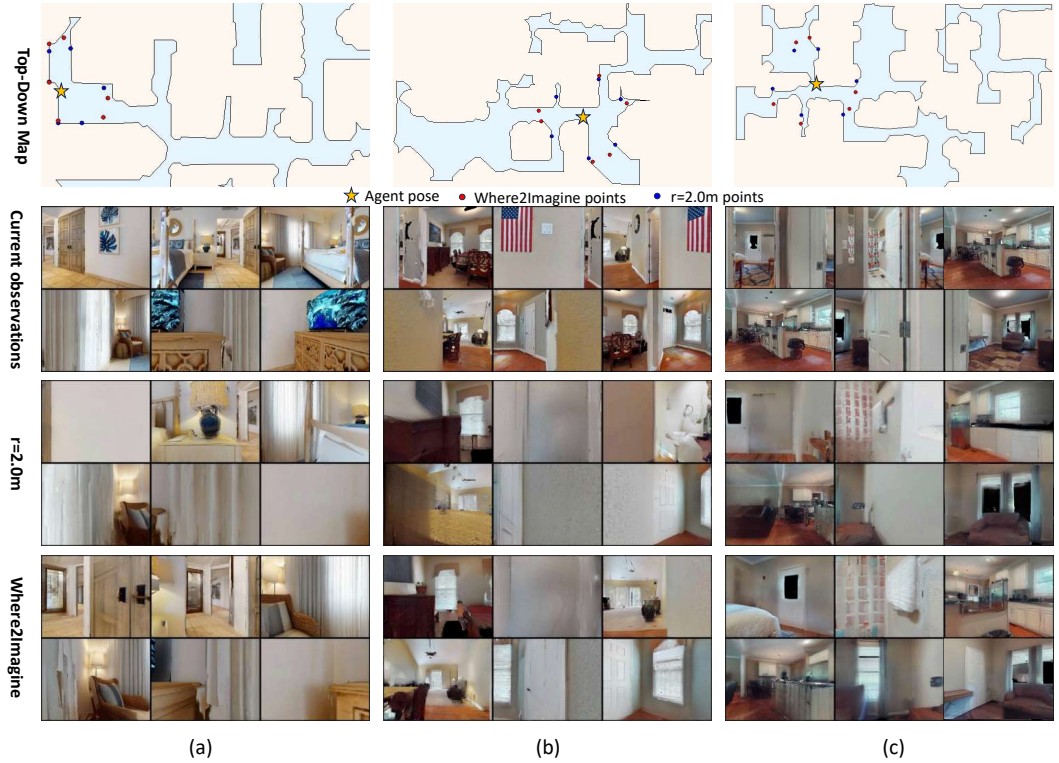

Figure 8: The visualization of the relative pose predicted by the Where2Imagine module and poses sampled at 60° intervals with 2.0m radius. The upper part shows the agent's current position (star marker) in different environments, as well as the distribution of the relative poses predicted by the Where2Imagine module (red dots) and the poses sampled at 60° intervals with a 2.0m radius (blue dots). The lower part shows the first-person views from different poses. Compared to uniform sampling at r=2.0m, the Where2Imagine module tends to predict more towards walkable areas and directions with higher information density.

### A.3   TRAINING DATASET FOR WHERE2IMAGINE

By replicating human demonstration trajectories, we collect first-person perspective images $I_t$ from the trajectory and, after T frames, use the relative camera pose $P_{t+T} = (\Delta x, \Delta y, \theta)$ as image labels to generate training data for the Where2Imagine module, as shown in Figure 9. After training, the module is able to predict the next waypoint based on current observations, aligning with human navigation preferences.

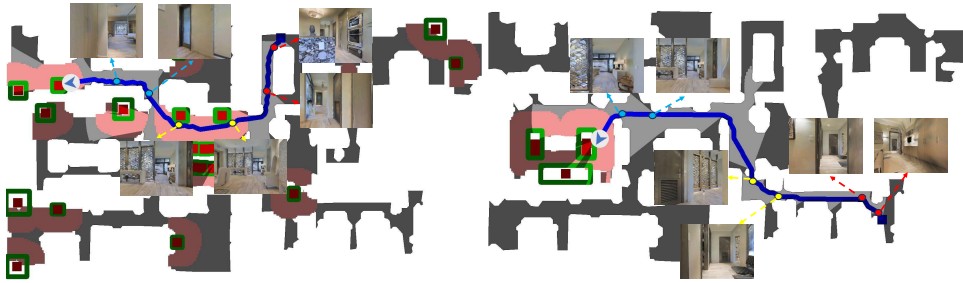

Figure 9: Visualization of human demonstration trajectories with different sampling intervals on the MP3D dataset in the Habitat-Web project. The left shows longer intervals, while the right shows relatively shorter intervals.

## A.4 SELF-CENTERED VISUALIZATION OF WHERE2IMAGINE

Figure 10 presents the prediction results of the Where2Imagine model on two different datasets, HM3D and HSSD. Each subfigure consists of four parts: the input image, the predicted waypoint (red dot), the observation image and the corresponding NVS image. These visualizations demonstrate the model's adaptability across different datasets and its ability to guide the agent towards more exploratory positions while avoiding obstacles.

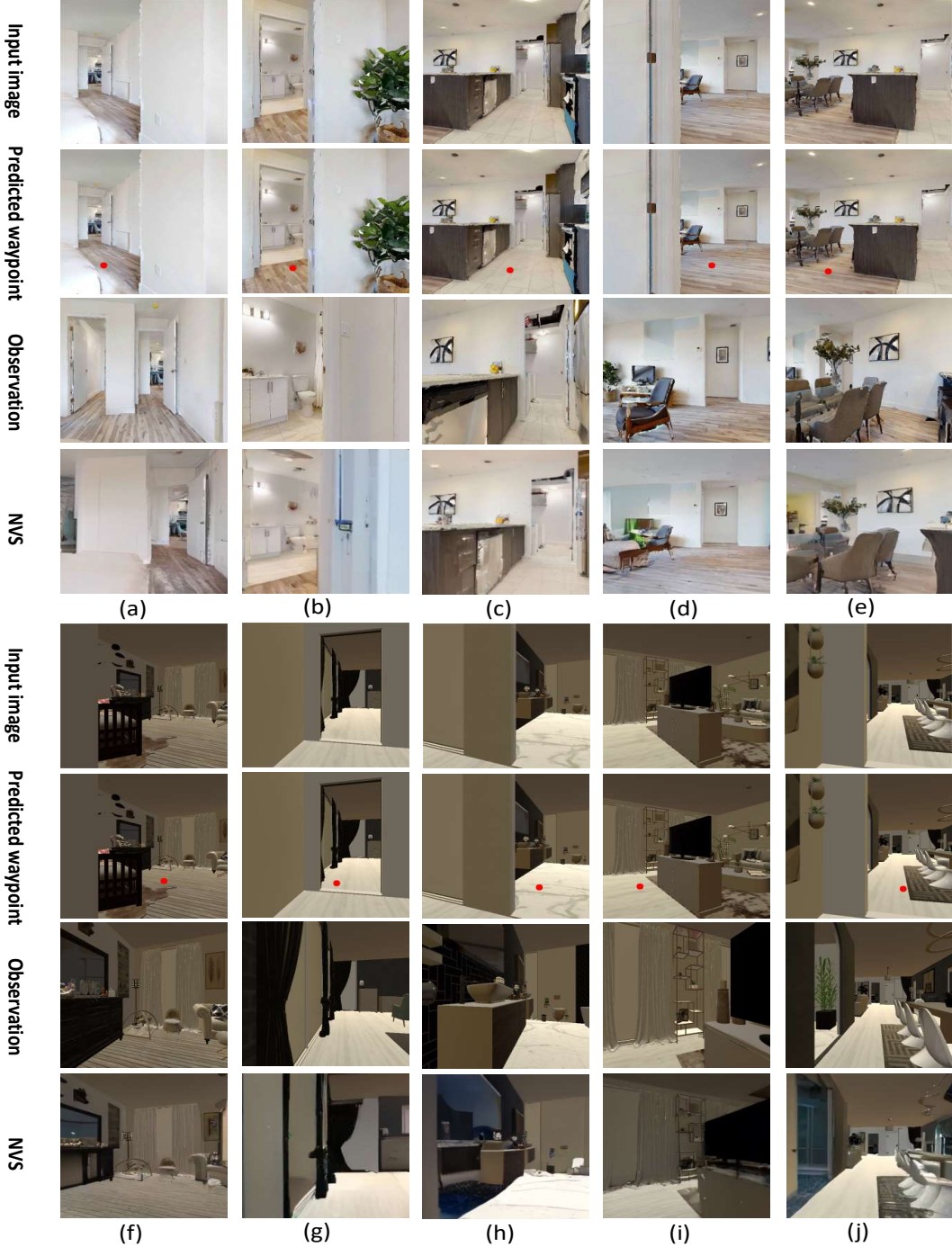

Figure 10: Self-centered visualization of the Where2Imagine model's prediction results. The top shows predictions on the HM3D dataset, while the bottom displays predictions on the HSSD dataset.