# OpenReview forum: "ImagineNav: Prompting Vision-Language Models as Embodied Navigator through Scene Imagination"
_ICLR.cc/2025/Conference — ICLR 2025 Poster_

### Official Review · Reviewer_Rhxu · 2024-11-04

**Soundness:** 3
**Presentation:** 3
**Contribution:** 3
**Rating:** 6
**Confidence:** 4

**Summary:**

This paper introduces ImagineNav, a novel framework that leverages VLMs for open-vocabulary object navigation without relying on explicit mapping or localization. The framework acts on a discrete action space spanning different views, then predicts candidate relative poses from the current observation, then uses NVS to generate images of those poses, and lastly uses a VLM for planning. The problem is posed as a best-view image selection problem for the VLM. Empirical experiments on challenging open-vocabulary object navigation benchmarks demonstrate that ImagineNav outperforms existing methods.

**Strengths:**

The paper is mostly clear and effective in conveying its points. There is an effective set of baselines indicating thorough experimental evidence but the addition of error bars would be helpful to determine significance. There are also effective sections discussing the failed and successful trajectories and clear ablations of the imagination module and the high-level planning module. The originality of having NVS methods apply to scene-level settings is interesting.

**Weaknesses:**

I think the main novelty within the method is the where2imagine module because prior work has used VLMs for high-level planning and NVS methods. I think the underlying claim is that waypoint as an intermediate representation (to predict poses and images with NVS) is a better reasoning intermediate than other intermediate representations and better than learning an end-to-end model. I think further investigating this claim could further add novelty to the paper as it would tackle a more fundamental question of are modular systems better than end-to-end models for object search and navigation. I would also try to rewrite the conclusion and portions of the introduction for clarity (paragraph 2). Lastly, the discussion of successful and failed trajectories are useful but I would like to see how to address those failed trajectories within the framework.

**Questions:**

Why do you think that waypoints are the correct intermediate representation for navigation and object search?
What is the point of having images from NVS as it limits performance and rather uses the embeddings to train a navigation policy?
What do you think are the limitations of the VLM high-level planning?
How can you get the VLM to learn online from additional data?

---

> ### Author Response · Authors · 2024-11-23
> **Discussions of using the point as subgoals and using hierachical policy.**
>
> Thank you for your kind suggestions and questions. From our perspectives, there are mainly three reasons for using points as intermediate subgoals instead of directly building end-to-end policies.
> (1) In real-world navigation scenarios, it is common for robots to interact with dynamic surroundings such as our humans. The action model must be able to run at high frequency, and it is difficult for an end-to-end VLA model to perform at high control speed.
>
> (2) The robustness of the point-goal navigation policy is confirmed in some recent works, such as iPlanner[1] and ViPlanner[2]. They propose efficient and lightweight models for collision avoidance and point-goal-reaching functions. It is much easier to build a general collision avoidance model than a navigation model to understand language task instructions. Therefore, the end-to-end model often suffers from poor generalization ability with limited data sources. During this rebuttal period, we evaluate the pre-trained SPOC [3] model, which is an end-to-end policy trained in ProcThor. The SPOC performs poorly in the HSSD and HM3D object navigation benchmarks.
>
> (3) The object navigation problem emphasizes the scene understanding for efficient exploration behaviors in novel environments. Such high-level knowledge for navigation task planning is already inherited in LLMs or VLMs trained with Internet datasets. It would be expensive and unnecessary to re-collect the embodied data with action labels to build the scene understanding abilities. But as the pre-trained VLMs tend to understand 2D images instead of 3D spaces, our proposed method provides an easy way to build a bridge between the VLMs to the point-goal navigation policy, which is imagination.
>
> We think one major disadvantage of using the VLMs for task planning is about agent memory. How to effectively encode the history information and let pre-trained VLMs to understand is one of the possible directions in the future works. And using in-context learning or RAG can help the VLMs to learn from additional navigation data.
>
> Reference:
> [1] Yang, Fan, et al. "Iplanner: Imperative path planning." Robotics, Science and Systems (RSS), 2023.
> [2] Roth, Pascal, et al. "Viplanner: Visual semantic imperative learning for local navigation." 2024 IEEE International Conference on Robotics and Automation (ICRA). IEEE, 2024.
> [3] Ehsani, Kiana, et al. "SPOC: Imitating Shortest Paths in Simulation Enables Effective Navigation and Manipulation in the Real World." Proceedings of the IEEE/CVF Conference on Computer Vision and Pattern Recognition. 2024.

---

> > ### Comment · Reviewer_Rhxu · 2024-11-26
> >
> > Thanks for addressing my concerns, I have marginally improved my score.

---

> > > ### Author Response · Authors · 2024-11-27
> > > **Official Comment by Authors**
> > >
> > > We are glad to hear that your concerns have been addressed. And thank you for your improving the score.

---

### Official Review · Reviewer_Va7n · 2024-11-04

**Soundness:** 3
**Presentation:** 2
**Contribution:** 3
**Rating:** 6
**Confidence:** 3

**Summary:**

This work presents an innovative framework for mapless visual navigation that utilizes Vision-Language Models (VLMs) to streamline open-vocabulary navigation. Unlike conventional approaches that depend on mapping and text-based planning, this method relies solely on RGB/RGB-D inputs, redefining navigation as a task of selecting the best view image. Using the Where2Imagine module, it generates potential viewpoints based on human-like navigation tendencies, enabling the VLM to select the most suitable view to locate target objects. The NVS module then generates potential views of the next waypoint. This approach results in efficient paths without mapping and enhances success rates on standard benchmarks.

**Strengths:**

1. This work introduces mapless navigation through the prediction of future 3D waypoints, generating possible image observations at these waypoints, and selecting the most promising next waypoint with VLM. The method is well-motivated and thoroughly described, making it feasible for the community to replicate the results.
2. The paper includes an extensive experimental evaluation of the HM3D and HSSD benchmarks, with the proposed method achieving notable improvements over baselines on the challenging HM3D and HSSD benchmarks.
3. A comprehensive ablation study is provided in Tables 2, 3, 4, and 5, highlighting the significance of different components within the proposed pipeline. Showing the effectiveness of using visual prompting and novel view thesis, as well as the waypoint prediction.
5. Figure 5 also offers some failure cases to help readers understand the method’s limitations.

**Weaknesses:**

1. The paper claims that object detection, segmentation, and mapping modules increase computational demand for robots. However, the proposed method introduces a computationally intensive novel view synthesis module to generate imagined observations. A comparison of computational load would strengthen this claim.

2. The paper’s organization could be improved for clarity. The text references to Tables and Figures are sometimes distant from the actual tables or figures; repositioning these elements could improve the flow and clarity of the paper.

3. Although GPT4o-mini is a robust VLM model, comparisons with recent open-source VLMs featuring 3D spatial understanding would enhance this work, such as:

* Spatial VLM: Endowing Vision-Language Models with Spatial Reasoning Capabilities

* SpatialRGPT: Grounded Spatial Reasoning in Vision Language Models

**Questions:**

1. While the provided failure case intuitively demonstrates a failure mode of the method, it would be valuable to include an approximate distribution of failure modes, such as how many failures are due to inaccurate imagined novel views.

---

> ### Author Response · Authors · 2024-11-23
> **We quantitively analyze the failure modes and append an additional experiment with SpatialRGPT.**
>
> Thank you for your valuable comments and suggestions. During this period, we quantitively analyze the failure modes in the HM3D benchmarks and append an additional experiment with SpatialRGPT. We find that the failures are mainly caused by the following reasons: 1) Stuck in the local optima. 2) Object detection errors. 3) GPT output errors. 4) Low Imagination Quality.
> The stuck-in-the-local optima error happens when the GPT 4-mini selects the explored directions. Object detection errors occur because of the rendered image quality in HM3D and HSSD. Sometimes, it tends to happen when similar objects appear simultaneously (sofa and chair). The GPT output errors are sometimes the hallucination happens and the GPT mistakenly thinks it has found the target objects and decides to stop another round of the decision process. The low imagination quality can sometimes lead to the wrong exploration direction and cause episode failure. But it is not the main reason.
>
> | Failure Reasons | Local Optima | Detection Error | GPT Response Error | Imagination Error | Others |
> |------------------|------------------|------------------|------------------|------------------|------------------|
> | HM3D (100 episodes) | 12%    | 19%   | 4%  | 3%  | 9%|
>
> We also follow your suggestions and conduct an ablation study using open-source SpatialRGPT to infer the direction. We use both perfect imagination and generated imagination to evaluate the planning ability of SpatialRGPT. The performance is shown in the table and we find that the SpatialRGPT can get similar performance on the HSSD dataset but gets confused using the HM3D-rendered images as inputs.
>
> | HM3D | Imagination | Success Rate | SPL |
> |------------------|------------------|------------------|------------------|
> |SpatialRGPT|Oracle|38.0|19.8|
> |SpatialRGPT|PolyOculus|35.0|16.2|
>
> | HSSD | Imagination | Success Rate | SPL |
> |------------------|------------------|------------------|------------------|
> |SpatialRGPT|Oracle|49.0|28.3|
> |SpatialRGPT|PolyOculus|45.0|24.3|
>
> Response to the concerns about computationally intensive novel view synthesis: Our method only needs to imagine the future observation at sparse timesteps, and on the contrary, the mapping and localization-based approaches require estimating the self-location and constructing the map in real-time. Although the current version of the novel-view-synthesis process still requires tens of seconds, but in the future, a lightweight model can further enhance the efficiency of our framework.

---

> ### Author Response · Authors · 2024-11-27
> **Thank you for your inspiring comments and questions.**
>
> Thank you for taking your precious time to review our paper. We are wondering whether the evaluation of different VLMs and the quantitive failure modes analysis help address your concerns. We are looking forward to your reply.

---

> > ### Comment · Reviewer_Va7n · 2024-12-03
> >
> > Thank you for addressing my concerns in the text and with additional experiments. I would like to keep my positive evaluations.

---

### Official Review · Reviewer_4kMy · 2024-11-05

**Soundness:** 3
**Presentation:** 2
**Contribution:** 2
**Rating:** 6
**Confidence:** 4

**Summary:**

This paper introduces a framework called ImagineNav, designed to enable effective visual navigation for robots in a mapless, open-vocabulary setting. ImagineNav leverages Vision-Language Models (VLMs) with on-board RGB/RGB-D camera inputs, without complex mapping and localization procedures. Instead of traditional planning approaches, it translates the navigation task into a series of "best-view" image selection problems, where the VLM selects optimal viewpoints based on imagined future observations. This approach views navigation as a simplified selection problem and showcases the potential of imagination-driven guidance for robotic systems.

**Strengths:**

1. This paper proposes a relatively novel navigation method: using novel view synthesis as a form of "imagination" for indoor navigation.
2. Due to the inclusion of a VLM and diffusion model, this work achieves a higher success rate in open-ended tasks.
3. The writing in this paper is clear, the illustrative figures are accurate, and the explanations of the proposed framework are well-articulated.

**Weaknesses:**

1. There are some concerns about the imagination module. The ImagineNav framework relies on the diffusion models for novel view synthesis. However, these could lead to some mistakes e.g. non-existing objects in the scene.  This could lead to incorrect navigation decisions by the VLM and reduce the overall success rate.
2. A small concern about this approach is its performance in multi-room or occluded scenarios. The use of human habits without interactive perception could cause the robot to become trapped in local optima, preventing it from locating the target.
3. The framework seems to only rely on immediate views for navigation decisions, without fully utilizing historical information from the navigation process. This lack of memory may limit the robot's ability to explore effectively and plan global paths over long distances or in intricate environments.

In summary, the method is novel (only to me, but if other reviewers show some related work, I will defer it). But the quality of the generated novel view (especially the emergence of non-existing objects) is concerned. If the author can explain more about this, I would increase the score.

**Questions:**

No extra problems.

---

> ### Author Response · Authors · 2024-11-23
> **Task planning with history experiment are added and more novel view synthesis visualization are appended.**
>
> Thank you for proposing your constructive concerns and questions. We try to deal with your concerns in the following content.
>
> Q1: **There are some concerns about the imagination module. The ImagineNav framework relies on the diffusion models for novel view synthesis. However, these could lead to some mistakes e.g. non-existing objects in the scene. This could lead to incorrect navigation decisions by the VLM and reduce the overall success rate.**
>
> Although the pre-trained novel view synthesis model cannot guarantee consistency in the test scenes, without additional in-domain fine-tuning, the imagined non-existent objects will not significantly influence the navigation performance. This is because imagination is only used to guide the VLMs selection process, and the object detection is still based on the real observed images. According to our experiment results, although the imagined images can get around 1/3 percentage (1331/4926) with non-existent objects, the failure episodes ratio caused by this is only 3/100. And we list all the failure reasons and their proportion in the table below. The most common failure reasons are still the detection error instead of the imagination error.
> | Failure Reasons | Local Optima | Detection Error | GPT Response Error | Imagination Error | Others |
> |------------------|------------------|------------------|------------------|------------------|------------------|
> | HM3D (100 episodes) | 12%    | 19%   | 4%  | 3%  | 9%|
>
> Q2: **A small concern about this approach is its performance in multi-room or occluded scenarios. The use of human habits without interactive perception could cause the robot to become trapped in local optima, preventing it from locating the target.**
>
> Thank you for raising these important concerns. We agree with the idea and how to gather the massive interactive exploration data to enhance the where2imagine module will be one of our future works.
>
> Q3: **The framework seems to only rely on immediate views for navigation decisions, without fully utilizing historical information from the navigation process. This lack of memory may limit the robot's ability to explore effectively and plan global paths over long distances or in intricate environments.**
>
> During the rebuttal period, we try to incorporate the history information in the planning process by concatenating a fixed number of history frames as additional inputs for VLM. Thanks for your suggestions and we find all the original methods can gain some improvements in both success rate and SPL. The comparison between the original approach with or without history frames is shown in the table below. We will calibrate all the metrics with history inputs in the paper tables after the rebuttal.
> | HM3D | Success Rate | SPL |
> |------------------|------------------|------------------|
> | ImagineNav (with history)    | 56.0    | 26.2  |
> | ImagineNav (without history)   | 56.0   | 24.3   |
>
> | HM3D | Success Rate | SPL |
> |------------------|------------------|------------------|
> | ImagineNav (with history)    | 55.0    | 28.1  |
> | ImagineNav (without history)   | 47.0 | 24.1 |
>
> Q4: **But the quality of the generated novel view (especially the emergence of non-existing objects) is concerned.**
>
> More visualizations are appended in the supplementary materials and as the statement in the answer of Q1, the non-existent objects from imagination will not significantly influence the performance as the object detection module still based on the real observation input to ensure whether the agent has found the final targets.

---

> > ### Author Response · Authors · 2024-11-27
> > **Thank you for your inspiring comments and questions.**
> >
> > Thank you for taking your precious time to review our paper. We are wondering if there are further concerns and whether the previous concerns are addressed. And we are happy to hear from your reply.

---

> > ### Comment · Reviewer_4kMy · 2024-11-27
> >
> > Thank you for your additional experiments and the detailed comments. I believe that the history one makes sense. But still, I am a little bit worried about the non-existing objects. Maybe in the future work that can be improved. However, I am impressed by the detailed results, so I will increase the score.

---

### Official Review · Reviewer_j4ao · 2024-11-06

**Soundness:** 2
**Presentation:** 2
**Contribution:** 2
**Rating:** 3
**Confidence:** 4

**Summary:**

This paper proposes "ImagineNav", a mapless navigation framework for robots using vision-language models (VLMs) to perform object search in open environments. It replaces traditional perception + mapping and text-based LLM-based planning with a pipeline where future scene views are "imagined" using novel view synthesis and analyzed by a VLM to choose optimal next waypoint. The candidate waypoints are generated by the Where2Imagine module, which is learned from human-like navigation patterns. Tested on benchmarks like HM3D and HSSD, ImagineNav significantly outperformed baselines in terms of success rate and success rate weighted by path length.

**Strengths:**

The system design significantly reduces complexity of an open-world object-goal navigation pipeline, and better integrates sensor observation with large models.

**Weaknesses:**

There are 3 fundamental limitations of this approach
- First, there is no way to guarantee that the diffusion model used to synthesize the novel views understands object permanence. How often does it hallucinate non-existing object or leave out objects that should be there? How does the rest of the pipeline deal with this?
- There is no way to ensure the waypoints generated by Where2Imagine is reachable, especially in out-of-distribution scenarios. The method still needs sine kind of local map for low-level path planning. Some tests on real robot or new environments will erase this concern.
- The method does not deal with the ambiguity of object reference. For example, navigate to the chair (example used in Fig. 2) is very ambiguous as there might be many chairs in the environment.

**Questions:**

- Some details are very unclear, especially on the novel view synthesis (i.e. future view imagination). What data is used to train the diffusion model? Is it in or out of distribution for indoor navigation? How about resolution and the speed?
- More qualitative visualizations of the environments, imagined views, waypoint distribution would be nice.
- The structure of the VLM analysis output in Fig. 3 and Fig. 4 are inconsistent. Which one is used in evaluation?
- Why is the success rate lower with NVS model added in Table. 2 (row 3 vs 5)? More explanation is needed.
- How does this method compare with end-to-end approaches like SPOC [1]?

[1] Ehsani, Kiana, et al. "SPOC: Imitating Shortest Paths in Simulation Enables Effective Navigation and Manipulation in the Real World." Proceedings of the IEEE/CVF Conference on Computer Vision and Pattern Recognition. 2024.

---

> ### Author Response · Authors · 2024-11-23
> **Qualitative visualizations and SPOC baseline performance are evaluated.**
>
> Thank you for raising the suggestions and concerns. And we try to answer your questions in the following content:
>
> Q1: **How often does it hallucinate non-existing object or leave out objects that should be there? How does the rest of the pipeline deal with this?**
>
> An efficient object exploration process should best utilize the prior knowledge about indoor scenes. As the novel-view synthesis modules are often trained with indoor scene datasets (such as RealEsate10k), the imagination module tends to imagine images that obey the training data. Although generated novel-view images may not ensure the object’s permanence, as our pipeline only imagines the future views at sparse timesteps, not every step, to provide additional information for VLMs planning, the imagined non-existent objects will not significantly influence the agent performance. We quantitively analyze 100 episodes process and visualize all the generated imagined images. We find there are 1331 images (4926 in total) containing non-existent objects compared with the label-rendered images in the scene. But the failure caused by this is only 3 episodes. This is because imagination is used as prior knowledge for exploration guidance, but an additional object detection module running on the real images is utilized to determine when and where to stop.
>
> Q2: **There is no way to ensure the waypoints generated by Where2Imagine is reachable, especially in out-of-distribution scenarios. The method still needs some kind of local map for low-level path planning. Some tests on real robot or new environments will erase this concern.**
>
> We quantify the success rate of the generated waypoint coordinates from the Where2Imagine module. Success is defined as the minimum distance between the waypoints to the scene obstacles is greater than 0.25m. We evaluate two versions of the Where2Imagine module, which differs in input modalities (RGB, Depth), and the performance is shown in Table below. All the evaluations are performed in the novel scenes, as the training data follows the Habitat-Web data, which uses MP3D as the data collection scenes. The evaluation scenes are HSSD and HM3D. We randomized the agent spawn pose 1000 times and let the Where2Imagine module predict the waypoints. Although the raw predicted waypoint can be unreachable, the low-level action predictor can deal with such scenarios.  And we will implement the real-world robot experiment after the rebuttal period.
> | Where2Imagine Input Modality | HSSD | HM3D |
> |------------------|------------------|------------------|
> | RGB   | 506/1000   |  370/1000  |
> | Depth | 545/1000   |  433/1000  |
>
> Q3: **The method does not deal with the ambiguity of object reference. For example, navigate to the chair (example used in Fig. 2) is very ambiguous as there might be many chairs in the environment.**
>
> Thank you for raising the important question. In this paper, we mainly discuss the object navigation problem, where a target object category is proposed as the goal. And find any of the chairs can satisfy the success criteria. But as the VLMs are flexible and we also use open-vocabulary object detections, our framework can be used for instance-level object searching.
>
> Q4: **Some details are very unclear, especially on the novel view synthesis (i.e. future view imagination). What data is used to train the diffusion model? Is it in or out of distribution for indoor navigation? How about resolution and the speed?**
>
> We use the pre-trained novel-view synthesis model proposed in PolyOculus: https://yorkucvil.github.io/PolyOculus-NVS/. No additional in-domain data are used to fine-tune the novel-view synthesis model. And the generated image resolution is 256x256. The generation speed of 6 images with a single RTX 4090 card is around 40 seconds.
>
> Q5 & Q6: **More qualitative visualizations of the environments, imagined views, and waypoint distribution would be nice. The structure of the VLM analysis output in Fig. 3 and Fig. 4 are inconsistent. Which one is used in evaluation?**
>
> Thank you for your kind suggestions. We append more visualization in the supplementary materials Figure 10. The Figure 10 contains both the visualization of the Where2Imagine output and the corresponding imagined images. In the experiments, we follow the VLMs output process shown in Figure 3, the Figure 4 is a summarized output from the output in Figure for illustration.
>
> Q7: **Why is the success rate lower with NVS model added in Table. 2 (row 3 vs 5)? More explanation is needed.**
> The row3 in Table 2 means that the agent uses the oracle imagination (the test scene renderer) and the row5 in Table 2 means that the agent uses a pre-trained novel-view-synthesis model. This gap can help evaluate how the imagination quality influences navigation performance. And we modify Table 2 to avoid possible ambiguity.

---

> > ### Author Response · Authors · 2024-11-23
> > **SPOC baselines comparision.**
> >
> > Q8: **How does this method compare with end-to-end approaches like SPOC?**
> > We evaluate the pre-trained SPOC model following the object navigation benchmark settings in HM3D and HSSD. We find that the SPOC achieves poor generalization in both benchmarks and report the metrics in the following table. Although the end-to-end navigation model is a more concise idea, the generalization across different domains still requires much more high-quality data to support.
> > | HM3D | Success Rate | SPL |
> > |------------------|------------------|------------------|
> > | SPOC   | 14.0    |  5.4  |
> >
> > | HSSD | Success Rate | SPL |
> > |------------------|------------------|------------------|
> > | SPOC   | 27.0    | 11.2    |

---

> > > ### Author Response · Authors · 2024-11-27
> > > **Thank you for your inspiring comments and questions.**
> > >
> > > Thank you for taking your precious time to review our paper. We are wondering if our appended experiment visualizations and end-to-end navigation model evaluation help address your concerns. And we are happy to hear from your reply.

---

### Author Response · Authors · 2024-11-23
**Thank you for all the reviewers.**

Thank you to all the reviewers raising crucial questions and suggestions for this paper. During the rebuttal period, we follow your suggestions and try to deal with your concerns from the following aspects:

(1) We quantitively analyze the failure reasons and visualize the ratio of different reasons, including the non-existence of objects issue caused by imagination in response to the Reviewer **j4ao , 4kMy and Va7n.** According to our analysis, although the imagination module can generate non-existent objects (at the ratio of 1331/4926 with 100 episodes), the ratio of failed episodes caused by this reason is only 3/100.  **As the agent stop flag is determined by object detection module on the real observed images,** the imagination of the non-existent object will not significantly influence the success rate.

(2) We conduct an additional experiment about introducing the history memory into the VLM inference process in response to the Reviewer **4kMy.** This is achieved by concating downsampled frames along the entire history RGB information and feed as an additional input for VLMs. We find that **including history RGB as input can help improve the SPL in both HM3D and HSSD** datasets (HM3D SPL from 0.243 to 0.262, HSSD SPL from 0.241 to 0.281), and significantly improve the ImagineNav performance on HSSD (Success Rate from 47% to 55%).

(3) We conduct an additional experiment about replacing the gpt4o-mini with the spatial understanding VLMs (SpatialRGPT) in response to the Reviewer **Va7n,** and we find the open-sourced SpatialRGPT **shows poor generalization to the HM3D domain rendered image data** and get lower than 40% success rate in the HM3D benchmark but got almost 50% percent success rate in HSSD.

(4) We quantify the quality of the Where2Imagine module waypoint prediction in response to Reviewer **j4ao**. We report the success rate for the generated waypoint (> 50%), where success is defined as the waypoint coordinates that lie on the obstacles. The metrics show that it is not difficult to get with pre-trained representations or using depth as input.

(5) We make a comparison between our method and the end-to-end method SPOC in response to the Reviewer **j4ao** and **Rhxu. We find that the end-to-end SPOC model,** although trained in massive simulation data, **shows limited generalization to the out-of-domain HSSD and HM3D object navigation** and achieved a success rate lower than 30% in both HSSD and HM3D benchmarks.

(6) We analyze the computation burden and the inference time of the novel-view generation module in response to the Reviewer **j4ao** and Reviewer **Va7n.**

(7) We improve the writing of the paper, add more detailed descriptions, and adjust the positions of Tables and Figures in response to Reviewer **Va7n.**

---

### Meta-Review · Area_Chair_Ld9b · 2024-12-21

**Metareview:**

The paper introduces ImagineNav, a novel mapless visual navigation system for robots using Vision-Language Models (VLMs) to guide object search. The approach replaces traditional mapping and localization with an imagination-driven method that generates and evaluates future views for navigation. The system outperforms existing methods on navigation benchmarks.

Strengths:

- The paper presents an novel method for navigation that reduces reliance on traditional mapping with a vision-language model-driven approach.

- Extensive experimental validation shows promising performance improvements.

- The reviewer appreciated the comprehensive ablation studies and failure analysis provided in the paper.

Weaknesses:

- The NVS module is computationally intensive, which might limit its use in real-time applications.

- There are concerns that the hallucination of non-existent objects in the imagined views could lead to incorrect navigation decisions.

- The lack of historical information in the navigation process limits long-distance planning, which was addressed by the authors in the rebuttal.

The paper presents a promising and innovative approach with promising experimental results. Despite concerns regarding scalability and hallucinated objects, most reviewers agree that the contributions are valuable. I recommend acceptance.

**Additional Comments On Reviewer Discussion:**

During the rebuttal, reviewers raised concerns about hallucinations of non-existent objects and the computational cost of the novel view synthesis (NVS) module. The authors addressed these by demonstrating that hallucinations had minimal impact on performance, as real object detection was used for decision-making. They also acknowledged the computational demands of NVS and suggested future improvements with a lighter model. Additionally, the authors incorporated historical navigation data, improving performance, and compared their method with other VLMs, showing promising results. These revisions led to a positive shift in reviewers' scores, supporting acceptance.

---

### Decision · Program_Chairs · 2025-01-22

Accept (Poster)